# Identification and Characterization of Copy Number Variations Regions in West African Taurine Cattle

**DOI:** 10.3390/ani12162130

**Published:** 2022-08-19

**Authors:** Félix Goyache, Lucía Pérez-Pardal, Iván Fernández, Amadou Traoré, Nuria A. Menéndez-Arias, Katherine D. Arias, Isabel Álvarez

**Affiliations:** 1Servicio Regional de Investigación y Desarrollo Agroalimentario, 33394 Gijón, Spain; 2CIBIO-InBIO, Universidade do Porto, 4485-661 Vairão, Portugal; 3Institut de l’Environnement et des Recherches Agricoles (INERA), Ouagadougou 04 BP 8645, Burkina Faso

**Keywords:** copy number variations, N’Dama cattle, Lagunaire cattle, *Bos taurus*, enrichment analysis, functional clustering, humid West Africa

## Abstract

**Simple Summary:**

Native West African (WA) taurine (Bos taurus) cattle are trypanotolerant; i.e., they have the ability to suppress the establishment of trypanosomes. Cattle trypanotolerance is a heritable trait—partially additive in nature—that can share mechanisms with tick resistance. Copy Number Variations (CNVs) are genomic alterations affecting gene structure, gene expression, and performance. However, the relationship between CNV regions and trypanotolerance and tick resistance in WA taurine cattle has not been approached thusfar. Here, we identify and characterize CNV regions in a sample of WA taurine cattle using WA zebu as an outgroup. The CNV regions identified spanned genes that are mainly involved in innate immunity. However, their relationship with genomic regions, previously reported to be associated with trypanotolerance and tick resistance, was not clear. Therefore, CNV alterations may not be the main genomic features underlying such traits.

**Abstract:**

A total of 106 West African taurine cattle belonging to the Lagunaire breed of Benin (33), the N’Dama population of Burkina Faso (48), and N’Dama cattle sampled in Congo (25) were analyzed for Copy Number Variations (CNVs) using the BovineHDBeadChip of Illumina and two different CNV calling programs: PennCNV and QuantiSNP. Furthermore, 89 West African zebu samples (Bororo cattle of Mali and Zebu Peul sampled in Benin and Burkina Faso) were used as an outgroup to ensure that analyses reflect the taurine cattle genomic background. Analyses identified 307 taurine-specific CNV regions (CNVRs), covering about 56 Mb on all bovine autosomes. Gene annotation enrichment analysis identified a total of 840 candidate genes on 168 taurine-specific CNVRs. Three different statistically significant functional term annotation clusters (from ACt1 to ACt3) involved in the immune function were identified: ACt1 includes genes encoding lipocalins, proteins involved in the modulation of immune response and allergy; ACt2 includes genes encoding coding B-box-type zinc finger proteins and butyrophilins, involved in innate immune processes; and Act3 includes genes encoding lectin receptors, involved in the inflammatory responses to pathogens and B- and T-cell differentiation. The overlap between taurine-specific CNVRs and QTL regions associated with trypanotolerant response and tick-resistance was relatively low, suggesting that the mechanisms underlying such traits may not be determined by CNV alterations. However, four taurine-specific CNVRs overlapped with QTL regions associated with both traits on BTA23, therefore suggesting that CNV alterations in major histocompatibility complex (MHC) genes can partially explain the existence of genetic mechanisms shared between trypanotolerance and tick resistance in cattle. This research contributes to the understanding of the genomic features of West African taurine cattle.

## 1. Introduction

Native West African taurine (*Bos taurus*) cattle are the result of a unique process of natural adaptation to the challenging environmental conditions of the biogeographic zones south of the Sahel, including the presence of different diseases, such as trypanosomiasis [1,2]. This adaptation process has formed different taurine cattle populations that are trypanotolerant [3].

Cattle trypanotolerance is a heritable trait consisting of the ability to suppress the establishment of trypanosomes in an individual and, in some cases, to reach self-recovery [4]. Since African animal trypanosomosis is a major obstacle to sustainable livestock production in sub-Saharan African countries [5], the ascertainment of the genetic basis of trypanotolerance is a major challenge in livestock genetics. Furthermore, there is evidence suggesting that the immune mechanisms involved in trypanotolerance are involved in resistance to other parasite infections as well [4]. In cattle, although *Bos indicus* is known to be more effective in counteracting tick infestation than *Bos taurus* [3], it has been reported that resistance to trypanosome infection can share mechanisms with resistance to tick attacks and tick-borne micro-organisms [6].

In this respect, trypanotolerant response has been experimentally shown to be primarily additive [7,8], with estimates of heritability varying from moderate to low [9]. Hanotte et al. [10], using an experimental herd, identified 40 significant trypanotolerance-related Quantitative Trait Loci (QTL) on 18 different bovine chromosomes (most of them located on BTA2, BTA4, BTA7, and BTA13). Although various genes have been proposed as candidates underlying some of these QTL, different studies have failed in identifying causal mutations in their sequences [11,12,13].

The QTL regions identified by Hanotte et al. [10] are too broad to be considered a precise map. Although these cattle chromosomal areas have been reanalyzed, increasing the set of markers used [9] and using selection sweep signals [14,15] or expression analyses [16], the refinement of these QTL regions is still an issue. In this respect, the information provided by genomic structural variants referred to as Copy Number Variations (CNVs) has not been approached thus far.

CNVs are DNA segments ranging in size from 50 base pairs (bp) to several megabases (Mb) in which insertion, duplication, or deletion events occur [17,18]. CNVs can affect gene structure and gene expression, and thus phenotypic variability [19]. There is consensus in considering CNVs as major genomic features associated with economically important traits and QTL in cattle [20,21].

The goal of the present research is to identify and further characterize CNV regions (CNVRs) in a sample of West African taurine cattle using West African zebu samples as an outgroup. Candidate genes located on the CNVRs identified were subject to functional characterization. The relationship between the candidate genes identified and QTL regions associated with cattle trypanotolerance and tick resistance was further assessed to contribute to a better understanding of the genomic features of native West African taurine cattle.

## 2. Materials and Methods

### 2.1. Samples, Genotyping, and Population Structure Analysis

Up to 106 West African taurine cattle samples were available and genotyped. Furthermore, 89 West African zebu samples were used as an outgroup. The taurine dataset included 33 Lagunaire cattle sampled in Benin (province of Mono); 48 N’Dama cattle sampled in the Comoé province of Burkina Faso (N’Dama (BF)); and 25 N’Dama cattle bred in Kolo (province of Bas-Congo), Congo (N’Dama (Co)). The zebu dataset included 21 Zebu Bororo cattle sampled in the provinces of Tahoua and Maradi of Niger, 12 Zebu Peul sampled in the Alibori province of Benin (Zebu Peul (Be)), and 56 Zebu Peul cattle sampled in the Dori Province of Burkina Faso (Zebu Peul (BF)). Previous descriptions of the populations sampled at the morphological, genetic, and breeding levels can be found in Traoré et al. [22,23], Moussa et al. [24], and Álvarez et al. [25].

Genotypes were obtained using the BovineHDBeadChip of Illumina (777,962 SNPs; (Illumina Inc., San Diego, CA, USA). The software GenomeStudio v2.0 (Illumina Inc., San Diego, CA, USA) was used to create both genotypic and intensity data useful for CNV calling and standard .ped and .map files useful for complementary analyses. SNPs were mapped on the bovine UMD 3.1 reference genome assembly using the NCBI genome remapping online service [26]. A total of 735,239 SNPs located on the 29 bovine autosomes were used.

Population structure was inferred via a standard multidimensional scaling (MDS) plot, as implemented in the program PLINK v1.9 [27], computed on a matrix of genome-wide pairwise identity-by-state distances. Before performing MDS, .ped and .map files were edited as follows: GenCall score cutoff was set to 0.15, mean sample call rate was set to 99%, and Minor Allele Frequency threshold was set to0.05. A total of 664,965 SNPs located on the 29 bovine autosomes were finally used. Coordinates computed on each individual were used to construct a bidimensional dispersion plot using the library ggplot2 [28] of R [29].

### 2.2. CNV Calling, Editing, and Candidate CNVRs

Following previous analyses in West African livestock [30], two different programs, namely PennCNV [31] and QuantiSNP [32], were used to perform CNV calling on the 29 autosomes of each individual in the dataset. Both software tools implemented Hidden Markov Models to detect CNVs based on the log of the observed probe hybridization intensity divided by the expected probe hybridization intensity of SNPs and the proportion of B alleles at an SNP. However, while PennCNV uses a transition matrix to model realistic copy number transitions between SNPs, QuantiSNP estimates posterior probabilities for copy number states for each SNP marker based on a Bayesian approach.

Results were edited as follows: (a) using PennCNV, three samples (1 N’Dama and 2 Zebu Peul individuals sampled in Burkina Faso) showing standard deviation for LRR > 0.3 or BAF drift > 0.01 were removed from the dataset and not used for further analyses; (b) using QuantiSNP, and following the authors’ recommendations, CNVs identified with a Log Bayes Factor < 10 were not considered informative and, therefore, were filtered out.

Potential CNV regions (CNVRs) were first constructed within cattle type (either taurine or zebu) using either PennCNV or QuantiSNP by merging CNVs overlapping by at least 1 bp across individuals using the merge function of the software BedTools [33]. The upper and lower bounds of these overlaps were as considered potential CNVRs for each cattle type. Since CNVs may reflect particular genomic scenarios at both the individual and the population levels [34], only potential CNVRs identified in at least three individuals or two populations (breeds) within each cattle type were considered representative of either taurine or zebu cattle and used to construct candidate CNVRs.

Finally, candidate CNVRs within cattle type were assessed using the intersect Bed function of the BedTools software and defined as the upper and lower bounds of the overlaps between potential CNVRs identified using each calling program. The CNVRs identified in each type of cattle were represented using the RIdeogram package [35] of R environment [29].

To ensure that candidate CNVRs reflect the genomic background of either native West African taurine (or zebu) cattle, candidate CNVRs which overlapped between cattle types were not used for enrichment analysis.

### 2.3. Enrichment and Functional Annotation Analyses

Using the BioMart tool [36], protein-coding genes spanned within the candidate CNVRs were retrieved from the Ensembl Genes 91 database, based on the bovine UMD 3.1 reference genome assembly. All the identified genes were processed using the functional annotation tool implemented in DAVID Bioinformatics resources 2021 [37] to determine enriched functional terms using the highest classification stringency. An enrichment score of 1.3, which is equivalent to a Fisher exact test *p*-value of 0.05, was used as a threshold to define the significantly enriched functional terms in comparison to the whole bovine reference genome background. Relationships among genomic features in different chromosome positions were represented using the software package shiny Circos [38].

### 2.4. Comparison with Previously Mapped QTL

Furthermore, previously reported QTL for trypanotolerance-related traits [10] and tick resistance were downloaded from the cattle QTL database [39,40] and overlapped with the candidate genes identified in taurine and zebu cattle. The QTL genome coordinates were re-mapped on the bovine UMD 3.1 reference genome assembly using the NCBI genome remapping online service [26] and overlapped with candidate genes using the BedTools software.

## 3. Results

The MDS plot in Figure 1 illustrates the dispersion of the bovine genotypes. The three taurine populations (negative values) are separated on the X-axis from the zebu individuals (positive values). The two N’Dama populations are separated from the other taurine cattle breed (Lagunaire) on the Y-axis. Zebu individuals showed high genomic homogeneity with very low dispersion across populations on both the X- and the Y-axis. Some N’Dama (BF) and, to a lower extent, Zebu Peul (BF) individuals took intermediate values on the X-axis, overlapping their dispersions.

### 3.1. Construction of Candidate CNV Regions

PennCNV allowed the identification of a total of 14,135 CNVs on all autosomes of 106 taurine and 87 zebu samples (Appendix A). In turn, QuantiSNP allowed us to identify 7997 CNVs on all autosomes of 105 taurine and 42 zebu individuals with Log Bayes Factor > 10 (Appendix A). Up to 2387 of the CNVs identified using QuantiSNP could be considered fully informative (maximum Log Bayes Factor > 30), while the remaining 5610 CNVs were considered suggestive (maximum Log Bayes Factor between 10 and 30).

Overlapping CNVs across individuals within type of cattle and calling program allowed us to construct potential CNVRs and to assess their representativeness within cattle type (i.e., identified in at least three individuals or two different populations within that type of cattle).

Regarding PennCNV, a total of 1929 potential CNVRs (of about 67 kb on average), covering roughly 130 Mb of the bovine genome, were identified in taurine cattle (Appendix A). Furthermore, a total of 1280 potential CNVRs (of about 44.5 kb on average), covering roughly 57 Mb of the bovine genome, were identified in zebu cattle (Appendix A). Only 722 and 433 of these potential CNVRs, covering 89.7 Mb and 38.1 Mb, respectively, were considered representative in taurine and zebu cattle, respectively.

Regarding QuantiSNP, a total of 1264 potential CNVRs (of about 79 kb on average), covering roughly 99.8 Mb of the bovine genome, were identified in taurine cattle (Appendix A). Furthermore, a total of 387 potential CNVRs (of about 73.5 kb on average), covering roughly 28.4 Mb of the bovine genome, were identified in zebu cattle (Appendix A). Only 504 and 157 of these potential CNVRs, covering 61.8 Mb and 18.9 Mb, respectively, were considered representative in taurine and zebu cattle, respectively.

Overlap between the potential CNVRs identified using the two calling programs allowed us to define a total of 388 candidate CNVRs of 180 kb on average in taurine cattle, covering about 70 Mb of the bovine genome (Appendix A). In zebu cattle, a total of 136 candidate CNVRs, of 165.2 kb on average and covering about 22.5 Mb of the bovine genome, were identified (Appendix A).

Overlapping among potential candidate CNVRs identified within cattle types allowed us to summarize them into 439 candidate CNVRs. Up to 306 of these candidate CNVRs were considered taurine-specific and 55 of them zebu-specific (Appendix A). Up to 78 candidate CNVRs were considered non-specific and, thus, were not used for enrichment analyses. Non-specific CNVRs were formed by the overlap of 82 taurine candidate CNVRs with 81 zebu candidate CNVRs (Figure 2A; Appendix A). The 307 taurine-specific CNVRs tended to be bigger (112 of them, 37.8%, sizing between 100 and 500 kb) than those identified in zebu cattle only (36 CNVRs, 65.4%, smaller than 50 kb length; Figure 2B). No zebu-specific CNVRs bigger than 1 Mb were identified (Figure 2C).

Figure 3 illustrates the distribution of the candidate CNVRs identified across bovine chromosomes per cattle type. Although taurine-specific candidate CNVRs were identified on all autosomes, zebu-specific CNVRs were not identified on BTA10, BTA16, BTA20, orBTA25. Moreover, taurine-specific CNVRs were uniformly distributed across autosomes, while most zebu-specific CNVRs were located on BTA3 (11%), BTA7 (9%), and BTA11 (9%). Although taurine and zebu cattle shared CNVRs on all bovine autosomes (Figure 2C; Appendix A), 51% of them were located on 6 bovine autosomes (BTA5, BTA6, BTA7, BTA12, BTA15, and BTA23).

### 3.2. Enrichment and Functional Annotation Analyses

Gene-annotation enrichment analysis allowed the identification of 840 candidate genes on 168 out of 307 taurine-specific CNVRs (Appendix A; 1527 different transcripts). For zebu-specific CNVRs, 64 candidate genes on 29 out of 54 CNVRs were found (Appendix A; 127 different 275 transcripts). These analyses gave a list of 64 candidate genes (Appendix A; 127 different transcripts) on 29 out of 54 zebu-specific CNVRs.

Functional annotation conducted on candidate genes allowed us to identify 41 different functional term annotation clusters in West African taurine cattle (Appendix A). However, only three of them (from ACt1 to ACt3) were statistically significantly enriched (enrichment score higher than 1.3). A description of these statistically significant functional terms is given in Table 1, and the gene relationships within annotation clusters are illustrated in Figure 4. ACt1 (enrichment score  =  2.27) included seven genes, most of them located on BTA11, belonging to the calycin/lipocalin proteins superfamily; ACt2 (enrichment score  =  1.47) included seven genes enriched for the splA/ryanodine (SPRY) receptor domain; and ACt3 (enrichment score  =  1.44) included nine genes belonging to the C-type lectin receptors family.

Twenty-five of the candidate genes identified belonged to the olfactory-receptor family (mainly located on BTA15 and 23; Appendix A). Interestingly, 196 out of the 840 candidate genes identified in taurine cattle were involved in one statistically non-significant cluster (ACt30; enrichment score = 0.175), enriched for membrane and transmembrane (Appendix A). ACt30 included some of the genes forming ACt1 (*PTGDS*), ACt2 (*LOC526787*), and ACt3 (*NKG2C, KLRK1, KLRD1, LOC100848575*, and *OLR1*).

Functional annotation conducted on the 64 zebu cattle candidate genes allowed the identification of two functional term clusters involved in olfactory-receptor activity and glycosylation (Appendix A). However, no annotation clusters identified in zebu cattle were significantly enriched (enrichment scores of 0.525 and 0.158, respectively).

### 3.3. Correspondence with Trypanotolerance-Related QTL

A total of 27 QTL previously reported for cattle trypanotolerance-related traits [10] overlapped with candidate genes spanning on 13 taurine-specific CNVRs on 8 cattle chromosomes (Table 2). Six of these CNVRs were located on BTA23. The 27 QTL were associated with 9 different traits mainly characterizing changes in body weight due to infection, changes in packed-cell volume, and parasite load. Candidate genes of five CNVRs (on BTA13, BTA20, BTA24, BTA26, and BTA27) overlapped with more than one QTL (from two to five). However, the candidate genes spanned on BTA23 overlapped with only one QTL (QTL_ID: 10543 associated with “Parasite detection rate”).

Four QTL regions associated with tick resistance reported by Machado et al. [42] and Otto et al. [43] on BTA23 overlapped with candidate genes currently identified in taurine cattle. Two other tick-resistance QTL identified in Nguni cattle by Mapholi et al. [44] overlapped with the *VAV2* gene on BTA11.

Interestingly, twenty-eight of the candidate genes spanned within four CNVRs identified on taurine cattle BTA23 overlapped with both trypanotolerance and tick resistance QTL traits [10,42,43] (Table 2). These genes, such as the major histocompatibility complex, class II gene (*BOLA-DRA*) and associated transcription factors (RNF39), and genes coding butyrophilin proteins (*BTNL2* and *LOC525599*), were mainly involved in immune function and olfactory-receptor activity (*OR2H1D*, *OR12D23*, *OR12D18*, *OR12D2H*, and *OR12D2E*).

Nine out of the thirteencandidate genes identified in zebu cattle overlapping with cattle trypanotolerance and tick resistance trait QTL were located on BTA23 (Appendix A).

## 4. Discussion

Most taurine and zebu samples used here were previously analyzed for population structure together with sanga cattle (ancient taurine X zebu crosses) using various statistical methods and both microsatellites and SNP arrays (see [15,45]). Consistently, Figure 1 confirms that (a) the samples analyzed are clearly structured due to the presence of two very different cattle types, zebu and taurine; (b) zebu individuals have a high genomic identity across populations; and (c) taurine samples include two different genetic backgrounds with a clear differentiation between the N’Dama and the Lagunaire cattle.

More important, Figure 1 confirms the importance of accounting for the existence of strong gene flow between West African cattle populations before carrying out analyses aimed at the ascertainment of unique genomic features within cattle type. A proportion of zebu and taurine individuals were intermingled and located in the central area of the dispersion plot. These individuals could be considered as sanga cattle in genomic terms [15,45]. Álvarez et al. [25] studied the gene flow patterns among neighboring cattle populations in Burkina Faso, suggesting introgression of Sahelian zebu genes into the Southern taurine cattle genetic background.

The current analysis used a significant number of zebu samples as a reference to ensure that the genomic features identified are unique for the West African taurine cattle genomic background. In this respect, a recent study reported that introgression due to between-population gene flow shapes the distribution of structural variations in cattle, allowing us to identify CNVs introgressed from zebu cattle in N’Dama cattle [46]. Therefore, the use of zebu samples as a reference is necessary to correctly identify CNVRs belonging to the native West African taurine cattle genomic background. In this respect, 26% and 63% of the 78 CNVRs shared by taurine and zebu cattle (Appendix A; Figure 2C) were defined using the information provided by individuals belonging to the five or six cattle populations analyzed, respectively. Furthermore, since the three taurine cattle populations analyzed are isolated by distance and have very different breeding histories, the conservative approach used to identify candidate CNVRs is likely to ensure that these CNVRs represent structural variations of importance within cattle type.

### 4.1. Identification of CNV Regions

The number and mean length of CNVRs identified in West African taurine cattle were significantly higher than in zebu cattle. Previous literature reported that CNV alterations are more frequent in *Bos indicus* than in African taurine cattle [34,47]. Therefore, the current results depart from expectations.

Studies focusing on CNVR identification are not directly comparable: they may differ due to the techniques used to obtain genomic data and both the genome assembly and number and performance of the calling platforms used [34,48,49]. Furthermore, population history and effective population sizes may cause differences in CNV abundance and mean CNV cumulative length between either different cattle groups or populations within groups. As examples, Asian zebu and African taurine cattle were reported to have higher CNV abundance than European cattle [50] and, within European cattle, British and Irish significantly differed in CNV counts from Balkan and Italian breeds [51].

In any case, studies reporting higher CNVR variability in zebu cattle used zebu samples of Asian or East African origin. Present East African zebu cattle are closely related to Asian zebu [47] due to a historically recent cattle restocking in this area using Asian zebu [52]. For that reason, the main source of discrepancy between the current and previous studies is likely to be due to the use of West African zebu samples that are considered a reservoir of ancient zebu introduction into Africa and have a completely different breeding history with intense gene flow with native African taurine cattle for millennia [25,52]. Since West African taurine (*Bos taurus*) and zebu (*Bos indicus*) cattle cannot be considered model representatives of their subspecies [52,53]; SNP array performance may differ from that of standard populations.

### 4.2. Biological Importance of the CNVRs Identified in Taurine Cattle

There is consensus that CNVs in cattle are highly enriched in the immune function and olfactory receptor genes, suggesting that Copy Number alterations contribute to their high variability [48,54]. This is why bovine chromosomes 15 and 23, respectively, have received particular attention in this kind of study [34].

The current results do not depart from expectations. Taurine-specific CNVRs identified on BTA15 are mainly enriched for genes coding olfactory receptors (Appendix A). Two CNVRs identified on taurine BTA23 are also enriched for this family of genes. Olfactory receptor genes are a gene family with a major function for ecological adaptation that has undergone extensive expansion and contraction through duplication and pseudogenization [55].

However, our results suggest that the candidate genes spanned by the taurine-specific CNVRs identified are mainly involved in innate immunity.

Genes belonging to ACt1 encode lipocalins, transport proteins delivering small hydrophobic ligands (fatty acids and, steroids) to cell surface receptors of specific cells. With highly diverse physiological functions, lipocalins are involved in the modulation of immune response and allergy [56]. Furthermore, lipid-processing protein, lipocalin 9 (LCN9) has been reported as a major candidate in conferring greater resistance of cattle to tick infestation [57].

ACt2 is formed by genes coding B-box-type zinc finger proteins, with most of them belonging to the tripartite motif (TRIM) encoding proteins and butyrophilins. TRIM proteins are upregulated by type I and type II interferons and they are well known to be involved in innate immune processes due to their role in the regulation of pathogenrecognition [58,59]. Interestingly, CNVRs identified in trypanotolerant Djalloné sheep have been recently reported to be enriched for this family of genes [30]. In turn, butyrophilin genes are part of the immunoglobulin genes superfamily, with immune system regulators being involved in different various infections in humans, such as tuberculosis [60].

Finally, ACt3 includes nine genes belonging to the C-type lectin receptors family, such as OLR1, involved in the immune response via regulating the inflammatory responses to pathogens and promoting B-cell differentiation and antigen-specific T-cell responses [61]. Carvalheiro et al. [62] suggested that the two genes of the regenerating islet-derived 3 family belonging to Cluster ACt3 (*REG3A* and *REG3G*) are candidates forplaying a role in regulating the cattle sensitivity to environmental variation under tropical conditions. The *REG3A* gene would be associated with wound repair after skin injury and with homeostasis of the skin, thus contributing to immune defense, whereas the *REG3G* gene would play a role in the antimicrobial defense of the intestine.

Other reasons contribute to relating taurine-specific CNVRs with immune function. CNV analyses in cattle paid major attention to BTA23, on which the major histocompatibility complex (MHC) genes are located. The major histocompatibility complex has a highly polymorphic nature [63], is well known to be involved in resistance to different diseases including gastrointestinal parasite resistance [64], and has been reported to be associated with antibody- and cell-mediated immune responses in cattle [65]. Taurine-specific CNVRs identified on BTA23 spanned candidate genes, such as *BOLA-DRA*, *GABBR1*, or *MOG*, belonging to MHC.

Furthermore, a high proportion of the candidate genes spanned by taurine-specific CNVRs belonged to a functional annotation cluster (ACt30) enriched for membranes and transmembranes. Although membrane and transmembrane genes represent a significant fraction (20–30%) of the total genes, at least in the human genome, their functionality is still poorly understood [66]. ACt30 includes a part of the genes belonging to the immune-related significantly enriched functional annotation clusters ACt1, ACt2, and ACt3, but also many other genes involved in immune functions, such as *BOLA-DRA*, the macrophage-stimulating 1 receptor (*MST1R*) gene, butyrophilin coding genes such as BTNL2, or the interferon-induced transmembrane protein 3 (*IFITM3*) gene, therefore suggesting that the main function of this set of genes may be related to immunity.

### 4.3. Relationship between Trypanotolerance-Related QTL

Although trypanotolerance is a heritable trait unique to West African taurine cattle, and the taurine-specific CNVRs identified are likely to be representative of their genomic background, the overlap between candidate genes spanned by these CNVRs and trypanotolerance-related QTL traits was not particularly high.

In principle, the existence of a high number of chromosomal regions putatively associated with cattle trypanotolerance [10] would fit well with the additive nature of the trait [7], assuming that these chromosomal areas are expected to have small to moderate effects on the trait. In any case, Hanotte et al. [10] used an experimental F2 population formed by crossing trypanotolerantN’Dama and trypanosuceptible Boran (East African zebu). QTL analysis of F2 populations may give highly biased QTL mapping [67], and therefore, the study of the chromosomal areas involved would need refinement [9,15,16]. Furthermore, although cattle trypanotolerance is partially determined by genes of additive nature, it is not expected that CNV alterations fully explain the additive variation of this trait.

In any case, the QTL regions associated with trypanotolerance may not be exclusive to regulating this trait. Some of the CNVRs identified on BTA23 of taurine cattle overlapped with the QTL regions reported by Machado et al. [42] and Otto et al. [43] for tick resistance using the same *Bos taurus* × *Bos indicus* experimental population. The general importance of BTA23 in the genetic basis of host resistance to parasites can be explained by the action of MHC genes including BOLA-DRA. This gene has been previously suggested to be under selection in West African cattle [2,15,68] and is a strong candidate forparticipating in resistance to other parasite attacks in an additive genetic manner. The existence of genetic mechanisms shared between trypanotolerance and tick resistance, as suggested by Mattioli et al. [6], could be based on CNV alterations.

Within CNVRs, our research failed at identifying the most proposed candidate genes for trypanotolerance in West African taurine cattle such as *CXCR4* [14] and *ARHGAP15* [16] on BTA2, *INHBA* [9] on BTA4, *TICAM1* [16] on BTA7 and, more recently, *CARD11* [69] on BTA25. A very recent association study in Baoulé cattle [70] reported two chromosomal regions on BTA16 and BTA24 potentially involved in trypanotolerance. The candidate genes spanning these genomic areas (*CFH*, *CRBN*, *TRNT1*, and *IL5RA*) were not identified within CNVRs either (Appendix A). Although trypanotolerance is likely to be controlled by a large number of loci of additive nature than by a limited number of major genes, our results do not preclude the possible action of these candidate genes on the trait. However, the mechanisms underlying their possible role may not be based on CNV alterations.

## 5. Conclusions

This research contributes to the understanding of the genomic features of West African taurine cattle. Our results suggest that the role of CNVs in the genomic history of these unique cattle is not negligible. Taurine cattle significantly differ in CNV variation of West African zebu. Furthermore, the identified CNVRs spanned genes that are mainly involved in innate immunity. This fits well with the expectation of a costly process of natural adaptation to the challenging environment of humid West Africa and its reflection on the cattle genome. The determination of the way in which CNVs have contributed to cattle trypanotolerance and resistance to parasites remains an issue. However, the study of CNVs on BTA23 appear to be a target for future research.

## Figures and Tables

**Figure 1 animals-12-02130-f001:**
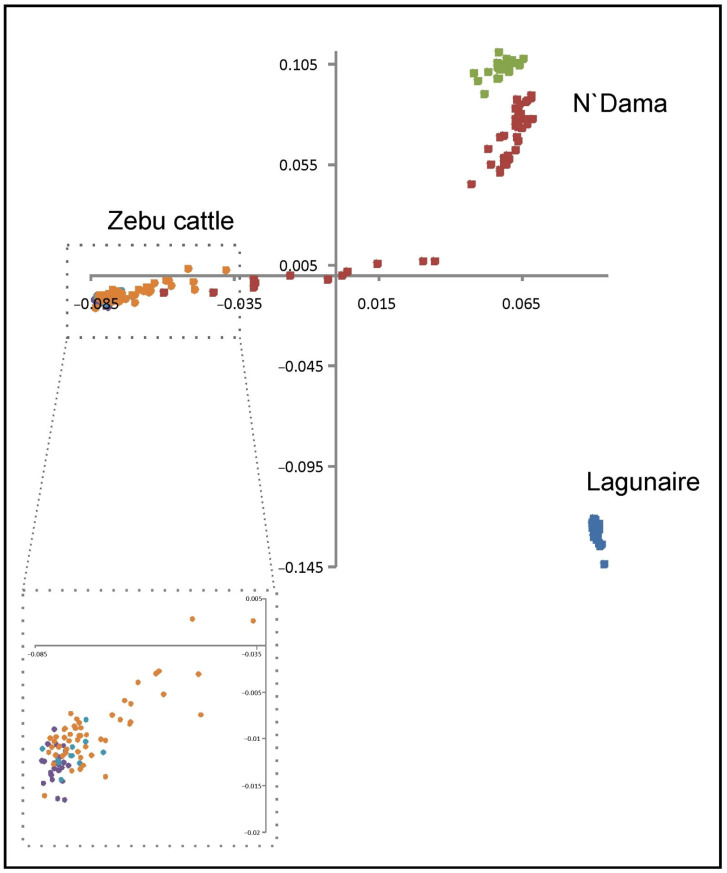
Bidimensional-scaling plot illustrating the dispersion of the West African taurine and zebu cattle genotypes. MDS was performed on the between-individuals identity-by-state distance matrix. Taurine cattle are in squares: light green, N’Dama of Congo; dark red, N’Dama of Burkina Faso; and blue, Lagunaire. Zebu cattle dispersion (in circles: blue, Bororo; purple, Zebu Peul of Benin; and orange, Zebu Peul of Burkina Faso) iszoomed in to illustrate the genomic homogeneity of these individuals.

**Figure 2 animals-12-02130-f002:**
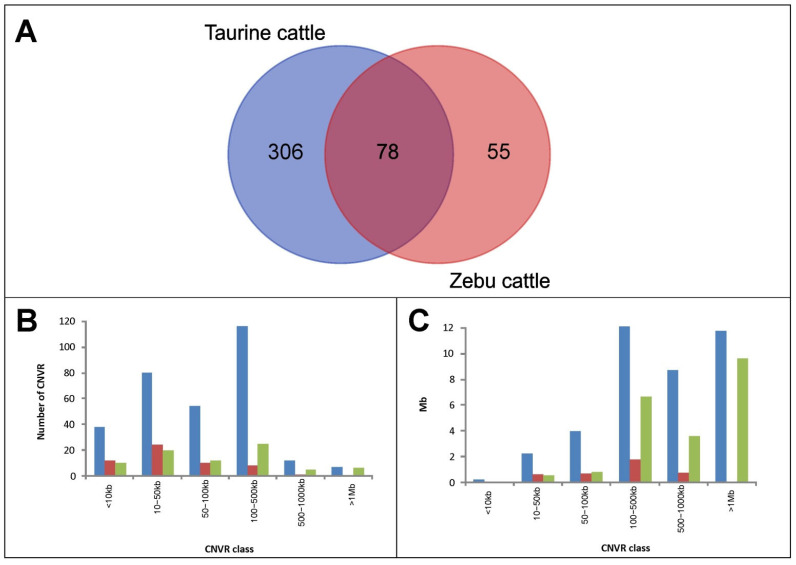
Venn diagram (Plot (**A**)) illustrating the number of candidate Copy Number Variation Regions (CNVRs) identified across type of cattle, constructed using an online tool [41]. Below, the histogram on the left (Plot (**B**)) illustrates the number of taurine-specific CNVRs (in blue), zebu-specific CNVRs (in red), and CNVRs shared between both cattle types (in green) per CNVR length class. Furthermore, the histogram on the right (Plot (**C**)) illustrates the total genome length (in Mb) covered by the CNVRs identified per type of cattle and CNVR length class.

**Figure 3 animals-12-02130-f003:**
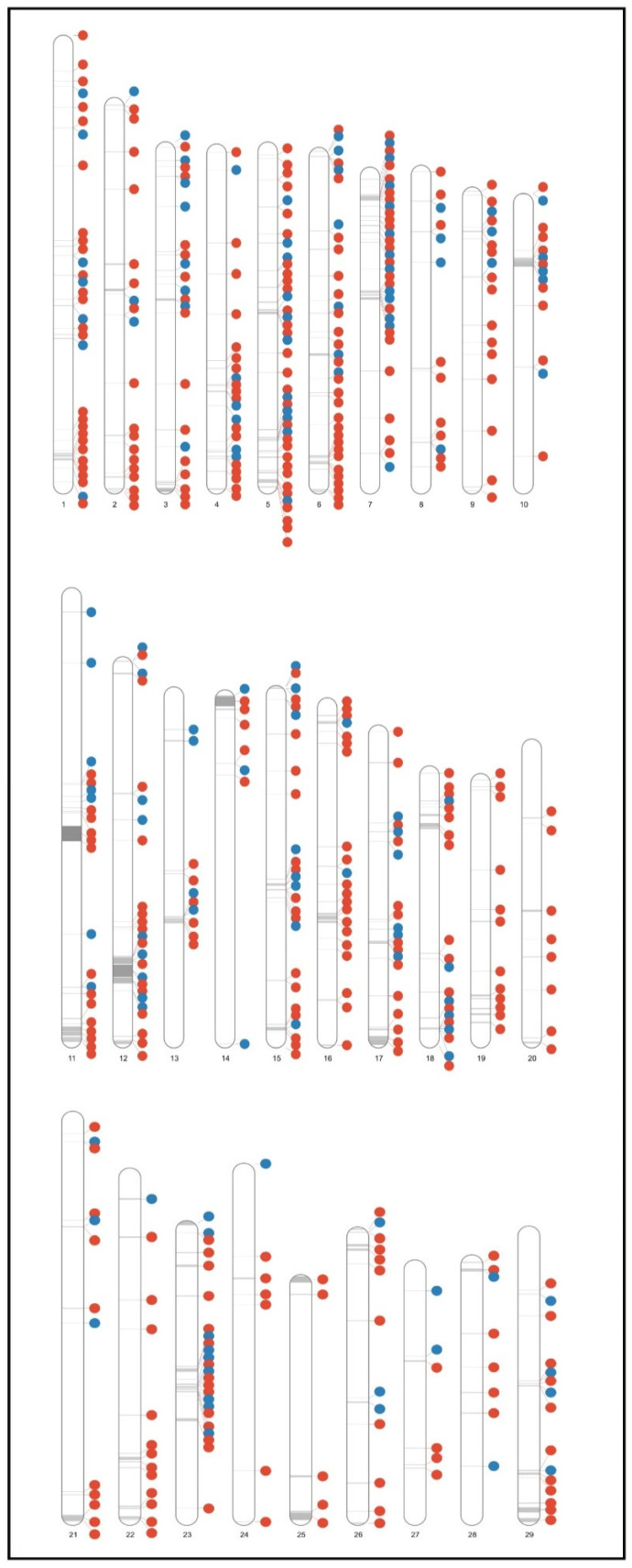
Ideogram illustrating, per bovine autosome, the distribution of the candidate taurine-specific (in red) and the zebu-specific (in blue) Copy Number Variations Regions (CNVRs) identified.

**Figure 4 animals-12-02130-f004:**
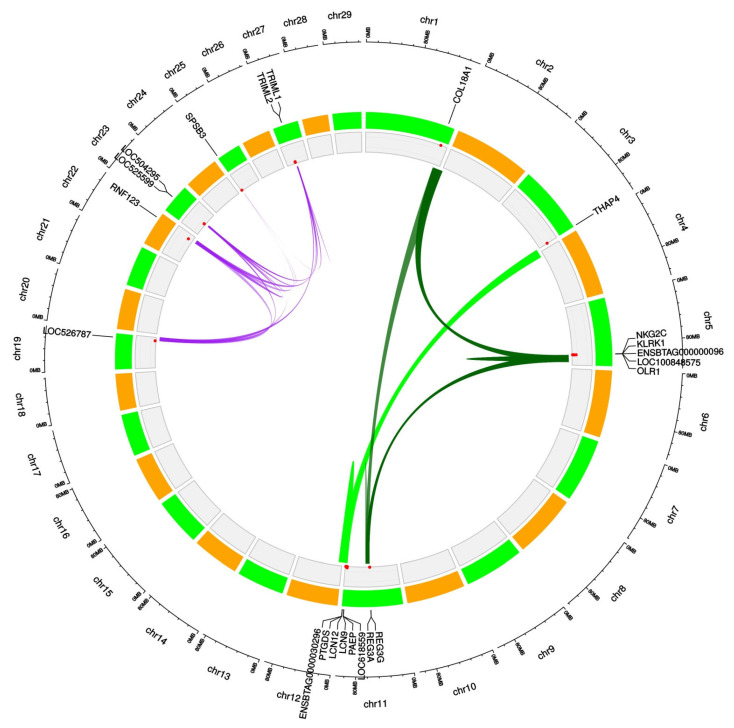
Circular map summarizing information on the three significantly enriched functional clusters identified in West African taurine cattle. Chromosomes are shown in the outermost circle. Candidate genes belonging to the significantly enriched functional annotation clusters are indicated next to their genomic localization. At the center of the map, links among candidate genes belonging to the same functional cluster are illustrated using the same color: ACt1 in light green; ACt2 in red; and ACt3 in dark green.

**Table 1 animals-12-02130-t001:** Significantly enriched functional annotation clusters and functional terms identified in West African taurine cattle.

Functional Cluster (Enrichment Score)	Category	Term and Description	Genes ^a^	*p*-Value	Fold Enrichment
Annotation Cluster 1 (2.27)	INTERPRO	IPR011038: Calycin-like	*THAP4, LOC618559, PAEP, LCN9, LCN12, PTGDS,* ENSBTAG00000030296	2.28 × 10^−3^	5.08
	INTERPRO	IPR000566: Lipocalin/cytosolic fatty-acid binding protein domain		7.28 × 10^−3^	4.85
	INTERPRO	IPR012674: Calycin		9.22 × 10^−3^	4.59
Annotation Cluster 2 (1.47)	SMART	SM00449: SPRY	*LOC526787, RNF123, LOC525599, LOC504295, SPSB3, TRIML2, TRIML1*	2.86 × 10^−2^	2.99
	INTERPRO	IPR003877: SPla/RYanodine receptor SPRY		0.031	2.96
	INTERPRO	IPR001870:B30.2/SPRY domain		0.044	2.71
Annotation Cluster 3 (1.44)	INTERPRO	IPR016186:C-type lectin-like	*COL18A1, NKG2C, KLRD1, KLRK1,* ENSBTAG00000000966, *LOC100848575, OLR1, REG3G, REG3A*	2.93× 10^−2^	2.47
	SMART	SM00034: CLECT		0.036	2.56
	INTERPRO	IPR001304:C-type lectin		3.98 × 10^−2^	2.51
	INTERPRO	IPR016187:C-type lectin fold		4.28 × 10^−2^	2.29

^a^ name of the gene retrieved from the Ensembl Genes 91 database. If only the Gene Stable ID was available, it is provided instead.

**Table 2 animals-12-02130-t002:** List of trypanotolerance-related and tick-resistance QTL overlapping with the candidate CNVRs identified in West African taurine cattle (CNVRt). The bovine chromosome (BTA) on which the QTL was identified; the position (in bp) for the start and end of the QTL; the QTL identification as obtained from the cattle QTL database [40]; the description of the trait associated with the QTL; and the candidate genes located in the CNVRs, including the functional annotation cluster (AC) they were assigned, are provided. Four CNVRs overlapping with both trypanotolerance and tick-resistance QTL are highlighted in bold.

Candidate CNVR	BTA	QTL Start (bp)	QTL End (bp)	QTL_ID	Trait	Candidate Genes
CNVRt4	1	12,484,941	34,327,730	10506	Initial packed red blood cell volume ^a^	*ROBO2* (AC17, AC30)
CNVRt195	11	104,664,393	104,664,433	101155	Tick resistance ^c^	*VAV2* (AC22, AC23)
				101169	Tick resistance ^c^	
CNVRt214	13	17,709,118	53,561,417	10524	Percentage decrease in PCV up to day 100 after challenge ^a^	*NET1* (AC23)
				10525	Percentage decrease in PCV up to day 100 after challenge ^a^	
				10526	Parasite detection rate ^a^	
CNVRt303	20	12,158,768	22,679,451	10537	PCVI minus PCVM ^a^	*ELOVL7* (AC30), *LOC107131564*, *DEPDC1B* (AC18)
				10538	PCV variance ^a^	
				10539	Percentage decrease in PCV up to day 100 after challenge ^a^	
				10540	Percentage decrease in body weight up today 150 after challenge ^a^	
				10541	Parasite natural logarithm of mean number ^a^	
CNVRt321	22	4,775,998	28,467,455	10542	Initial packed red blood cell volume ^a^	*SCN5A* (AC30)
CNVRt334	23	4,584,848	29,554,995	10543	Parasite detection rate ^a^	*FAM83B*
CNVRt335	23	4,584,848	29,554,995	10543	Parasite detection rate ^a^	*BAK1*, *LOC516410*, *ITPR3* (AC23, AC30), *MNF1*, *IP6K3* (AC30), *LEMD2* (AC30), *MLN*
**CNVRt336**	23	4,584,848	29,554,995	10543	Parasite detection rate ^a^	*KCNK5* (AC30, AC5), *KCNK17* (AC30, AC5)
		13,002,524	13,489,143	164932	Tick resistance ^d^	
		13,002,524	13,489,143	164933	Tick resistance ^d^	
**CNVRt338**	23	4,584,848	29,554,995	10543	Parasite detection rate ^a^	ENSBTAG00000048364, ENSBTAG00000038397, ENSBTAG00000015565 (AC30), *BOLA*-*DRA* (AC30), *BTNL2* (AC30), *LOC525599* (AC2, AC30), *LOC504295* (AC2), ENSBTAG00000026163, ENSBTAG00000050817
		22,432,428	29,658,279	12223	Tick resistance ^b^	
**CNVRt341**	23	4,584,848	29,554,995	10543	Parasite detection rate ^a^	ENSBTAG00000054439, *RNF39*, *PPP1R11*, *POLR1H* (AC31), ENSBTAG00000054588, *ZFP57* (AC37), *MOG* (AC30), *GABBR1*(AC30, AC8), ENSBTAG00000031825, *LOC504548*, ENSBTAG00000052703, *OR2H1D* (AC30, AC40,AC41), *OR2H1* (AC30,AC40,AC41)
		22,432,428	29,658,279	12223	Tick resistance ^b^	
		28,511,975	39,129,003	12224	Tick resistance ^b^	
**CNVRt342**	23	4,584,848	29,554,995	10543	Parasite detection rate ^a^	*OR12D23*, *OR12D18*, *OR12D2H*, *OR12D2E* (AC30, AC40, AC41)
		22,432,428	29,658,279	12223	Tick resistance ^b^	
		28,511,975	39,129,003	12224	Tick resistance ^b^	
CNVRt345	23	28,511,975	39,129,003	12224	Tick resistance ^b^	*ZKSCAN8* (AC31, AC37)
CNVRt346	23	28,511,975	39,129,003	12224	Tick resistance ^b^	ENSBTAG00000038430, *PRP4* (AC9), *PRP8* (AC9), *PRP6* (AC9), *PRP1*, LOC100298767 (AC9), *PRP2*
CNVRt349	24	14,587,194	22,401,178	10544	Initial packed red blood cell volume ^a^	ENSBTAG00000044087
				10544	Percentage decrease in PCV up to day 100 after challenge ^a^	
CNVRt364	26	5,469,941	18,747,171	10548	Percentage decrease in PCV up to day 100 after challenge ^a^	*LOC540627*
				10549	Percentage decrease in PCV up to day 100 after challenge ^a^	
				10550	BWF scaled by BWI ^a^	
				10551	Body weight (mean) ^a^	
				10552	Percentage decrease in body weight up today 150 after challenge ^a^	
CNVRt369	27	13,701,246	20,999,044	10553	PCVI minus PCVM ^a^	*ZFP42* (AC37), *TRIML2* (AC2), *TRIML1* (AC2)
				10554	PCV variance ^a^	
				10555	Percentage decrease in PCV up to day 100 after challenge ^a^	
				10556	Percentage decrease in PCV up to day 100 after challenge ^a^	

^a^ QTL reported by Hanotte et al. [10]; ^b^ QTL reported by Machado et al. [42]; ^c^ QTL reported by Mapholi et al. [44]; ^d^ QTL reported by Otto et al. [43].

## Data Availability

The data presented in this study are available from the corresponding author upon reasonable request.

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
