# Peer review of "Identification and Characterization of Copy Number Variations Regions in West African Taurine Cattle"

_animals, 2022, doi:10.3390/ani12162130_

Round 1
Reviewer 1 Report
This is an interesting topic and a well-written manuscript.
It is unclear why CNV was investigated in this case. It would be appreciated if the authors were to provide more background on why CNV was thought to be involved or if this was a fishing expeditions to see if CNV could be causal.
The results are interesting. Have you considered expanding your analysis beyond protein-coding genes to examine transcription factor binding sites?
Author Response
Reviewer 1
Comments and Suggestions for Authors
This is an interesting topic and a well-written manuscript.
ANSWER: Thank you very much
It is unclear why CNV was investigated in this case. It would be appreciated if the authors were to provide more background on why CNV was thought to be involved or if this was a fishing expeditions to see if CNV could be causal.
ANSWER: To our knowledge, this is the first time that West African cattle populations have been assessed for CNV. This paper belongs to a project including a series of works aiming to contribute to the ascertainment of the genomic basis of cattle trypanotolerance. Considering both that CNV are known to be involved in the phenotypic variation of quantitative traits and that trypanotolerance has a partially additive genetic basis, the assessment of CNV variation in West African cattle was, in our honest opinion, necessary. Anyway, we are not proud enough to consider that our findings do not need further verification and refinement.
The results are interesting. Have you considered expanding your analysis beyond protein-coding genes to examine transcription factor binding sites?
ANSWER: our project focuses on sequence variation. At present we do not have evidence enough to determine a target genomic region in which the analyses suggested by Reviewer 1 should be performed.
Reviewer 2 Report
The authors have presented a study investigating CNVs in several West African taurine and indicine cattle breeds, looked briefly at the population structure of these breeds using MDS, and compared the CNVRs identified to known genes for trypanotolerance and tick resistance. The manuscript is well written and mostly quite clear; however, I have a few queries and comments:
Why have the authors used the UMD 3.1 genome reference instead of the much newer (but still several year old) ARS1.2 reference? The latter is much better assembled (especially in gene-rich regions with high GC content in my experience), so using the new reference when identifying CNVs should give more accurate results. It would also enable the authors to use more up-to-date gene annotations: Ensembl 91 is from 2017, and has fewer genes annotated (less accurately) than the latest version (107). The choice is especially surprising as the authors state that they used a remapping service, presumably from the newer reference to the older one.
Figure 1: taurine cattle (N'Dama and Lagunaire) are shown with squares (not circles as described in the figure caption); also, Zebu cattle are shown with circles (not triangles). In addition, the colours used in the figure don't match the descriptions in the caption: N'Dama are shown in red and green, not blue and light green. Lagunaire are shown in blue, not olive green. I can't tell if the colours are accurate for the Zebu breeds.
Line 148: which annotation sets were included in the DAVID analysis? What classification stringency? Also, DAVID 6.8 is quite old (from 2016) and has now been retired.
Line 177, 178: "allowed to identify" should be either "allowed us to identify" or "allowed the identification of". Similar for line 184 with "construct" and "construction"
Line 192: what do you mean by CNVRs being considered representative? Does this mean breed-specific?
Figure 3: chromosome number labels are too small to read without zooming in a long way
PAEP (in annotation cluster 1) encodes β-lactoglobulin, a major milk protein: are any other genes in this cluster related to milk production? Lipocalins involved in fatty acid transport sound like plausible candidates, since milk contains a lot of fat. Are any of the taurine or zebu breeds sampled in the study used as dairy cattle? I believe that Butyrophilin (in CNVRt338) is also associated with fat droplet production in milk.
Line 444: is there any reason to assume that that these mechanisms would be based on CNVs? Protein coding variants or regulatory changes caused by promoter variants would seem at least as likely to me.
Author Response
Reviewer 2
The authors have presented a study investigating CNVs in several West African taurine and indicine cattle breeds, looked briefly at the population structure of these breeds using MDS, and compared the CNVRs identified to known genes for trypanotolerance and tick resistance. The manuscript is well written and mostly quite clear; however, I have a few queries and comments:
ANSWER: Thank you very much
Why have the authors used the UMD 3.1 genome reference instead of the much newer (but still several year old) ARS1.2 reference? The latter is much better assembled (especially in gene-rich regions with high GC content in my experience), so using the new reference when identifying CNVs should give more accurate results. It would also enable the authors to use more up-to-date gene annotations: Ensembl 91 is from 2017, and has fewer genes annotated (less accurately) than the latest version (107). The choice is especially surprising as the authors state that they used a remapping service, presumably from the newer reference to the older one.
ANSWER: Reviewer is right. However, this paper belongs to a project including a series of works aiming to contribute to the ascertainment of the genomic basis of cattle trypanotolerance. The other published papers belonging to this project were performed using the UMD 3.1 genome reference. Therefore, for consistency, the present research was carried out using such reference genome.
Figure 1: taurine cattle (N'Dama and Lagunaire) are shown with squares (not circles as described in the figure caption); also, Zebu cattle are shown with circles (not triangles). In addition, the colours used in the figure don't match the descriptions in the caption: N'Dama are shown in red and green, not blue and light green. Lagunaire are shown in blue, not olive green. I can't tell if the colours are accurate for the Zebu breeds.
ANSWER: thank you for pointing out this error. Figure caption has been amended.
Line 148: which annotation sets were included in the DAVID analysis? What classification stringency? Also, DAVID 6.8 is quite old (from 2016) and has now been retired.
ANSWER: thank you for pointing out this error. The version used was DAVID 2021. We used the highest classification stringency. These changes have been included in the text of the manuscript.
Line 177, 178: "allowed to identify" should be either "allowed us to identify" or "allowed the identification of". Similar for line 184 with "construct" and "construction"
ANSWER: when necessary, lines were rephrased as suggested throughout the text.
Line 192: what do you mean by CNVRs being considered representative? Does this mean breed-specific?
ANSWER: this line is now rephrased as “….representative of either taurine or zebu cattle…”
Figure 3: chromosome number labels are too small to read without zooming in a long way
ANSWER: sorry, but we could not make these numbers bigger.
PAEP (in annotation cluster 1) encodes β-lactoglobulin, a major milk protein: are any other genes in this cluster related to milk production? Lipocalins involved in fatty acid transport sound like plausible candidates, since milk contains a lot of fat. Are any of the taurine or zebu breeds sampled in the study used as dairy cattle? I believe that Butyrophilin (in CNVRt338) is also associated with fat droplet production in milk.
ANSWER: the taurine West African cattle populations analyzed were never used for dairy and, therefore, such interpretation may exceed the presented evidence that, in turn, fits well with the hypothesis underlying the work. In any case, we agree with Reviewer 2 that many genes in the cattle genome may have pleiotropic effects, as we have suggested in other works belonging to that project.
Line 444: is there any reason to assume that that these mechanisms would be based on CNVs? Protein coding variants or regulatory changes caused by promoter variants would seem at least as likely to me.
ANSWER: We agree with Reviewer 2. CNV may be one of few mechanisms underlying that performance. However, since the traits explored are known to have, at least partially, an additive genetic basis and considering that CNV are involved in the phenotypic variation of quantitative traits, the assessment of CNV variation can contribute to the knowledge of the genomic basis of adaptation of West African cattle.